# Causes of Vaccine Hesitancy in Adults for the Influenza and COVID-19 Vaccines: A Systematic Literature Review

**DOI:** 10.3390/vaccines10091518

**Published:** 2022-09-13

**Authors:** Simran Kumar, Zayna Shah, Sara Garfield

**Affiliations:** 1University College London School of Pharmacy, London WC1N 1AX, UK; 2Imperial College Healthcare NHS Trust, London W6 8RF, UK; 3NIHR Imperial Patient Safety Translational Research Centre, London SW7 2AZ, UK

**Keywords:** vaccine hesitancy, adults, influenza, COVID-19

## Abstract

Background: Vaccine hesitancy was labelled as one of the top ten threats to global health by the World Health Organization in 2019 and is associated with negative health outcomes. Previous reviews on cause of vaccines have not included vaccine hesitancy related to the COVID-19 vaccine. This review aimed to fill this gap by synthesising the findings of studies identifying causes of vaccine hesitancy to the COVID-19 and influenza vaccines. Methods: A systematic literature review was conducted. Searches were carried out in the PubMed, EMBASE and Cochrane databases. Following data extraction, a thematic analysis was conducted of the causes of vaccine hesitancy in adults for the influenza and COVID-19 vaccines. Results: Fourteen papers were included. Four themes were identified as causes of vaccine hesitancy comprising: concerns over safety, lack of trust, lack of need for vaccination and cultural reasons. While concerns over safety were found in all countries, some of these were specific to particular countries and cultures. Our findings suggest that scientific knowledge of vaccines and size of clinical trials during their development reduce vaccine hesitancy. However, pharmaceutical companies were not a trusted source of information. Conclusion: Our findings build on those of previous research to suggest specific information that may be helpful in addressing vaccine hesitancy. Targeted approaches from trusted sources are needed to address specific safety concerns.

## 1. Introduction

Vaccine hesitancy has been identified as one of the top ten threats to global health by the World Health Organization (WHO) due to the risk of reducing progress made in terms of controlling vaccine-preventable diseases [1]. The WHO has also identified that between two and three million deaths are avoided per annum due to vaccination [1]. The 2019–2020 influenza vaccination programme is thought to have prevented 7.5 million influenza infections, 3.7 million influenza-related medical appointments, 105,000 influenza-related hospitalisations and 6300 influenza-related deaths [2]. However, a study that modelled the effects of vaccine hesitancy found that high vaccine hesitancy could result in a 7.6 times higher mortality rate over a two-year period in the absence of non-pharmaceutical interventions [3]. It also found that high levels of vaccine hesitancy increased the need for non-pharmaceutical interventions [3].

Whilst studies have been conducted in adults in countries throughout the world on their reasons for vaccine hesitancy, few studies have provided a global perspective of this issue. A recent systematic review and thematic analysis synthesised factors affecting hesitancy to the Swine flu and Ebola vaccines [4]. A key limitation identified by the authors was that the review did not include studies relating to Corona Virus Disease-19 (COVID-19) and that it was important to explore how the findings of the review related to the COVID-19 pandemic [4]. This review aimed to fill this gap by synthesising the findings of studies identifying causes of vaccine hesitancy to the COVID-19 and influenza vaccines.

By focussing on vaccines that are recommended for large populations, studies focussing on large cohorts around the world with differing socioeconomic backgrounds were included in the review, to provide a global synthesis of the causes of vaccine hesitancy in adults.

## 2. Methodology

### 2.1. Search Strategy

A systematic review of papers addressing the causes of vaccine hesitancy in adults was carried out by searching the PubMed, EMBASE and Cochrane databases using the keywords (reasons OR causes) AND (vaccine hesitancy OR refusal) AND (adult) (see Appendix A for example). We checked our search strategy against the PRESS Peer Review of Electronic Search Strategies: 2015 Guideline Statement [5].

Peer-reviewed studies carried out in the last 20 years identifying reasons for vaccine hesitancy in relation to the influenza and COVID-19 vaccines in adults were included. Studies that only focussed on a specific age group, condition or profession were excluded, as were studies specifically exploring vaccine hesitancy in pregnancy. Studies were also excluded if they were not written in English, or did not define an adult as being at least 18 years of age or older. Reviews, editorials, conference abstracts or letters were also excluded. Studies focussing on an unlicensed/unavailable vaccination (for example, studies from the start of the pandemic which focussed on attitudes towards a potential COVID-19 vaccine) were excluded.

### 2.2. Screening, Reliability Checks and Risk of Bias Assessment

The results from the database searches were combined and duplicates removed. All titles and abstracts were screened by one reviewer (SK), and 10% were independently screened by a second reviewer (ZS) (the agreement level was 96%). All full texts were screened by SK, and a 10% sample was independently screened by ZS (agreement level 93%). All discrepancies were resolved via discussion to obtain a final agreement level of 100% at each stage. Data were extracted from the studies including country, study design sample size, population, type of vaccine, prevalence of vaccine hesitancy and causes of vaccine hesitancy. ZS independently extracted data a 10% sample of the papers (agreement level was 100%).

Cross-sectional survey studies were assessed for risk of bias using the AXIS appraisal tool for cross-sectional studies. Focus group studies were appraised using the Critical Appraisal Skills Programme (CASP) checklist for qualitative research. A 10% sample of the papers was appraised independently by a ZS for reliability of the quality assessment phase (agreement level 90%). All discrepancies were resolved via discussion to obtain a final agreement level of 100%.

A meta-analysis was not appropriate due the heterogeneity of the data, and a narrative synthesis of the data was carried out using thematic synthesis. Causes of vaccine hesitancy were coded for each study, and similar codes between studies were then merged to form four overarching themes.

## 3. Results

In total, fourteen papers met the inclusion criteria (Figure 1).

### 3.1. Study Characteristics

The study characteristics for the 14 included studies are presented in Table 1 (see Appendix A for full data extraction tables). Of the 14 studies, 5 focused on the influenza vaccine and 9 focussed on the COVID-19 vaccine. The studies were carried out in Somalia, Pakistan, USA, Jordan, India, Italy, Hungary, Canada and South Africa, with one study focusing on those of Arabian ethnicity from 23 Arab countries and territories and 122 other countries.

### 3.2. Risk of Bias in the Included Studies

Having evaluated the studies using the CASP and AXIS tools, it was found that in many studies, there was a degree of bias related to participant selection where this was skewed towards certain populations; for example, a study which recruited respondents for their online survey via a snowball sampling strategy where the online survey was distributed via social media and media platforms of a university found their sample to be skewed towards students [6]. In addition, there was limited information on non-responders in most of the studies. However, the results of studies with less representative samples were consistent with studies that had representative samples. In addition, although most studies did not mention potential bias during the formulation of the questions included in the questionnaires, studies [9,12,14,16] established good internal reliability.

### 3.3. Causes of Vaccine Hesitancy

Four themes were identified in relation to vaccine hesitancy: cultural reasons, lack of need for vaccination, concerns over vaccine safety and lack of trust. These are described further below. Where quantitative information was available, this is stated as a percentage of all participants of the study, rather than only those who were vaccine hesitant, unless otherwise stated.

#### 3.3.1. Concerns over Vaccine Safety

Concerns over vaccine safety vaccination were identified in the vast majority of included studies. Participants were concerned about side-effects ranging from minor side effects to concerns about potential undisclosed side effects that would occur post-immunisation (61.4%) [14]. Six percent of participants in one study reported that they were concerned due to an adverse reaction to a previous dose of the vaccination [16]. There was hesitancy associated with the COVID-19 vaccine in particular in terms of safety. Many participants underestimated the size of clinical trials with only 15% of respondents (95% CI: 12.2% to 17.7%) correctly identifying that the Pfizer/BioNTech and Moderna clinical trials combined had over 50,000 participants [11]. Those who were not vaccine hesitant were significantly more likely to select the largest trial size estimates than those who were unvaccinated (33.8% vs. 21.0%; d, *p* < 0.01) [11].

There was concern that both the coronavirus and influenza vaccines contain live COVID-19 [11] and influenza strains [18] and that these can cause COVID-19 and influenza, respectively, in addition to the concern that the COVID-19 vaccine could harm those with lower levels of immunity or with comorbidities [12]. Those who were vaccine hesitant were significantly more likely to believe that the vaccine contained live coronavirus (54.5% vs. 35.3% *p* < 0.001, two-tailed test) [11]. For the influenza vaccine specifically, there were concerns about not knowing enough about the vaccine (Soweto: 10, 8%; Klerksdorp: 9, 10%) and not wanting to become habituated to it [19].

There were concerns surrounding the safety of the manufacturing process as participants in some lower income countries (including one-third of participants in a study in Jordan) believed that COVID-19 vaccines manufactured in the USA or Europe were the safest [9] and some participants feared receiving a faulty or fake vaccine [12]. Believing in conspiracy theories was an independent factor in predicting vaccine acceptance; if respondents believed in conspiracy theories, then they were less likely to accept vaccination (OR = 0.502, 95CI% = 0.356–0.709, *p* < 0.001) [9]. In addition, some African Americans identified the Tuskegee Syphilis Study as an explanation for their concerns over becoming vaccinated [14]. Interestingly, one study showed that 61.2% of vaccine hesitant participants changed their opinion or agreed to reconsider this after a six-month period [17]. Their intent to change their mind was linked to the availability of reliable information on the safety and adverse effects of vaccination from the government [17].

#### 3.3.2. Lack of Trust

Lack of trust was another theme identified in the vast majority of included studies and whilst there were specific concerns regarding the COVID-19 vaccine, there was a considerable lack of trust in both the influenza and COVID-19 vaccines. For the COVID-19 vaccine in particular, a lack of trust in the government and belief in conspiracy theories played a large role in vaccine hesitancy [13], especially in those who were vaccine resistant (87.6%) [17].

There was a lack of trust in the COVID-19 vaccine, as it had been newly developed (58.7% vs. 55.5% Bologna vs. Palermo) [13], and discussion of vaccine passports had made participants distrust their government as they did not want to be forced to be vaccinated (25.4% Bologna) [13].

Whilst participants acknowledged that social media and friends/family were not reliable sources of information [9], less than 60% of respondents believed in the ability of pharmaceutical companies to make a safe and effective vaccine, and many did not have trust in their government to be able to provide the vaccine for free [9]. A lack of trust in healthcare policies (39.1%), pharmaceutical companies and published studies (33%) caused high levels of vaccine hesitancy [15], as did a lack of information [15]—a study conducted on participants of Arabian ethnicity found that those who were unaware of the types of COVID-19 vaccines available to them were more likely to be vaccine hesitant (OR = 1.93, 95% CI = 1.82–2.06), as were participants who felt that they did not have enough information/clarity about the vaccine, its safety and adverse effects [15].

#### 3.3.3. Lack of Need for Vaccination

Respondents reported a variety of beliefs about the need for vaccination. Some were of the view that it was unnecessary as they rarely contracted infectious diseases (67.7%) [19], the vaccine would be ineffective (6.22%) [10] or that their immune system was sufficient to handle the infection (4.49%) [10]. A lack of knowledge of both vaccinations and the diseases they prevented presented a challenge, as 30% of participants believed that influenza cannot cause death [19]. In places where the influenza vaccine was provided for free, some participants were hesitant to be vaccinated as they felt that the vaccine being free does not guarantee anything [16].

Participants were more likely to have a vaccination administered if they felt it was necessary to protect others (r = 0.574, *p* < 0.001) [8]. For example, participants in one town in one study were significantly more willing to receive the influenza vaccine if a member of their household was HIV-positive and thus more susceptible to severe illness from the disease (aRR 0.3, 95% CI 0.1–0.8) [19]. There were some COVID-19-specific beliefs, such as that the vaccine may not be effective against new strains (20%) [15] or that there was no need for it as the case load in the respondent’s country was decreasing (identified from focus group study) [12]. In addition, some participants expressed the view that it was natural to have the flu, thus mitigating the need to be vaccinated against it [19].

Some of the beliefs reported concerning lack of necessity of vaccines were linked to that of cultural reasons. For example, the belief that vaccination was unnecessary due to trust in God was significantly associated with vaccine hesitancy (AOR 2.45; 95% CI 1.34–4.48) [7]. Additionally, belief in lack of necessity can be used to explain racial differences in vaccine uptake as a study showed that White Americans had a higher perceived risk from influenza than African Americans, and this may have contributed to lower influenza vaccine uptake in the latter ethnic group [15]. Interestingly, many (48.9%) participants believed that they did not have a risk factor for influenza, thus vaccination was unwarranted; however, upon further analysis, over half of these participants had at least one risk factor for influenza [10]. In addition, a study investigating the effects of racial fairness on vaccine hesitancy found that respondents who perceived there to be higher levels of racial fairness, were significantly more likely to agree on their moral obligation to get vaccinated to protect others (*p* < 0.05) [14]. There was also a non-significant trend towards lower use of natural remedies amongst those who perceived there to be higher levels of racial fairness (which was important as African Americans had a statistically significant higher dependency on naturalism as an alternative to vaccination and a lower level of knowledge on the vaccine) [14]. A study investigating the relationship between influenza vaccine and race found that discrimination increased vaccine hesitancy, as could feeling unfairly treated by a healthcare professional and that racial fairness significantly (*p* < 0.05) impacted a multitude of positive attitudes and beliefs regarding the flu vaccine in both White and African Americans [14].

#### 3.3.4. Cultural Reasons

Cultural reasons for vaccine hesitancy were identified in Somalia, Pakistan, South Africa and Malaysia [6,7,15,18,19]. These reasons included hesitancy to have the COVID-19 vaccine due to fears of it containing substances from pork and fears of it causing sterility [6]. The fear of sterility meant that females were less likely to accept the COVID-19 vaccine as one study in some African settings found that the ability to procreate is seen as a mark of womanhood [6]. In this study, female gender was significantly associated with vaccine hesitancy [6]. Other reasons included belief in natural remedies and prayer as an alternative to vaccination, with trust in God and prayer being significantly associated with vaccine hesitancy (AOR 2.45; 95% CI 1.34–4.48) [19]. One study found that Buddhist respondents were twice less likely to accept the COVID-19 vaccine as Islamic respondents [18].

## 4. Discussion

This review has identified four main themes underpinning the reasons behind vaccine hesitancy in adults in relation to the influenza and COVID-19 vaccines. These are concerns over safety, lack of trust, lack of need for vaccination and cultural reasons. There were specific reasons for COVID-19 vaccine hesitancy and influenza vaccine hesitancy such as perceived rushed development of the vaccine [11] and not wanting to become habituated to an annual vaccine [19], respectively.

This review has demonstrated that similar factors affecting parent’s decisions to vaccinate their young children [20] also affect adults’ decisions about vaccinations. Smith et al. [20] carried out a systematic review and found that the safety of vaccines was the most frequently reported reason for vaccine hesitancy. In addition, Smith et al. [20] found that belief in lack of susceptibility to an illness seemed more important than belief in lack of belief about severity, a finding also echoed by our review.

In addition, our study has found that specific factors affecting decisions about accepting vaccines during previous pandemics [4] have also affected decisions to be vaccinated against COVID-19. These include concerns about safety and efficacy of vaccines linked to the speed of development of the vaccine, lack of trust in governments and misinformation. Our study has additionally found that mistrust in pharmaceutical companies also had a role. Belief in personal susceptibility was an important factor both in previous pandemics [4] and the COVID-19 pandemic.

### 4.1. Recommendations for Practice

The findings of our review build on previous reviews in being able to suggest a way forwards in tackling the causes of vaccine hesitancy. Our review findings suggest that there is potential to change vaccine hesitancy by providing information. This builds on the findings of a systematic review and meta-analysis of 29 studies regarding the willingness of parents and guardians to vaccinate their children against COVID-19 that found that access to scientific information [21] was significantly associated with vaccine acceptance. Our review further suggests that important information to include is the nature of vaccines (particularly if no live vaccine is given) and the numbers of people included in clinical trials. This information needs to come from trusted sources, rather than from pharmaceutical companies directly. Religious and cultural beliefs were one cause of concerns regarding vaccinations. This suggests religious leaders may be an important vehicle for providing information in some settings. Our review suggests that concerns over safety are common across geographical locations but that specific concerns may be more common in some geographical locations and cultures than others. A targeted approach is therefore needed. In addition, Smith et al. [20] had found that parents did not think that the protection of other children was a reason to vaccinate their own. However, our review shows that where people perceived that someone in their own household was at risk, they were more likely to be vaccinated. This suggests that information regarding the protection of others may be more effective if it is tailored to the protection of close contacts.

While it is known that some COVID-19 vaccines have very rare serious side effects [22], it was the fear of unknown side effects or side effects that people had heard of through non-medical sources that seemed to be a cause of concern for participants. Helping individuals evaluate the actual risks and benefits of vaccination and make their own informed decisions may therefore be an important part of addressing vaccine hesitancy. Psychological approaches to this have been suggested in previous literature [22].

### 4.2. Strengths and Limitations

The strengths of this study include the range of countries the included studies were carried out in, allowing the study to have a global perspective on causes of vaccine hesitancy. Many of the included studies had large sample sizes. Limitations include the exclusion of studies not written in English being included. In addition, this review did not include studies which only focussed on certain sub-groups of the population (e.g., teachers or military personnel). Future work could potentially explore if such subgroups displayed a higher degree of vaccine hesitancy than the general population to allow for recommendations for interventions targeted to these subgroups to be made. The literature on vaccine hesitancy in relation to COVID-19 vaccines is rapidly growing, and future reviews are likely to identify more studies.

## 5. Conclusions

Vaccine hesitancy has been described as one of the top ten threats to health by the World Health Organisation (WHO) [1], and much research has been carried out in the area in the causes of vaccine hesitancy in adults. Our review findings taken in combination with other reviews, suggest that concerns about vaccines and beliefs about lack of personal susceptibility to threats are common causes of hesitancy to vaccines. Such concerns may be exacerbated during pandemics due to the speed of vaccine development. Targeted approaches to specific concerns from trusted sources are needed to address this.

## Figures and Tables

**Figure 1 vaccines-10-01518-f001:**
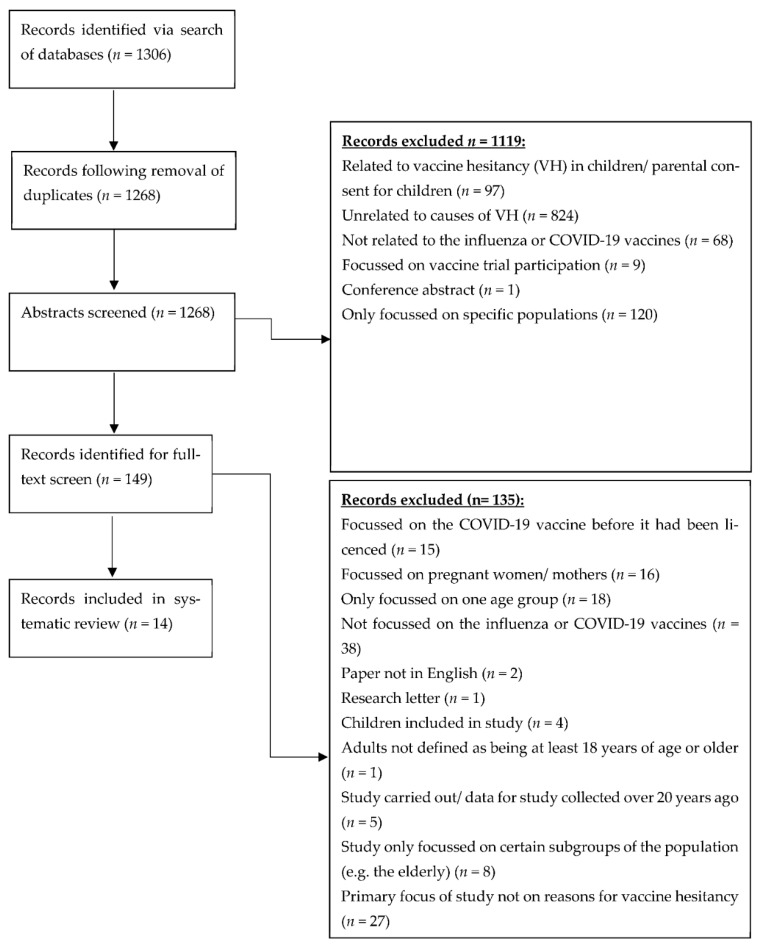
PRISMA diagram of papers identified, screened and included.

**Table 1 vaccines-10-01518-t001:** Study characteristics of the final 14 studies.

Study	Methods	Participants	Causes of Vaccine Hesitancy
Ahmed et al. [6]	Cross-sectional survey	>18 years, residing in Somalia (*n* = 4543)	Cultural reasons, not needed (6.22% of respondents felt it was ineffective) and it being dangerous (9.33% of respondents were scared of side effects)
Ahmed et al. [7]	Cross-sectional survey	>18 years, residing in Pakistan. (*n* = 655)	Not needed (belief that a Muslim’s trust in God is enough protection was significantly associated with vaccine hesitancy (AOR 2.45; 95% CI 1.34–4.48), vaccination is dangerous, lack of trust and cultural reasons
Dorman, et al. [8]	Cross-sectional survey	>18 years, residing in Orange County, USA, (*n* = 26,324)	Not needed, vaccination is dangerous (confidence in vaccination safety was a key determinant of willingness to be vaccinated (r = 0.723, *p* < 0.001))
El-Elimat et al. [9]	Cross-sectional survey	>18 years, residing in Jordan (*n* = 3100)	Lack of trust, vaccination is dangerous (<60% respondents believed that pharmaceutical companies would be able to make a safe and effective vaccination; 49.6% reported that they would not have the vaccine due to side effects)
Galistianiet et al. [10]	Cross-sectional survey	Aged 20–59 years, residing in Hungary (*n* = 1631)	Not needed (55.4% of unvaccinated participants did not believe that influenza vaccination was the best way to prevent influenza); vaccination is dangerous
Kreps et al. [11]	Cross-sectional survey	>18 years and above. (*n* = 1027)	Not needed; vaccination is dangerous (63.9% of the hesitant respondents thought the side effects would be severe)
Kumari et al. [12]	Thematic analysis of focus group discussions	>18 years, residing in India (*n* = 39).	Not needed, dangerous, lack of trust (the study findings suggested that trust in the safety of vaccines was a driver for a positive attitude towards vaccine acceptance)
Montalti, et al. [13]	Cross-sectional survey	>18 years old from Bologna and Palermo (*n* = 443)	Not needed, dangerous, lack of trust (24.4% of respondents in one city cited they were aware of cases where people had become “damaged” as a result of vaccination)
Quinn et al. [14]	Cross-sectional survey	819 African American 838 White respondents, all >18 years	Not needed (African Americans had a statistically significant higher dependency on naturalism as an alternative to vaccination), dangerous, lack of trust
Qunaibi et al. [15]	Cross-sectional survey	Adults of Arab ethnicity from 145 countries (*n* = 36,220)	Not needed, dangerous (55.7% of respondents had concerns about the safety of the vaccine), lack of trust
Roy et al. [16]	Cross-sectional study	>18 years in US (*n* = 108,700)	Not needed (66–74% respondents felt it was not necessary), dangerous, lack of trust
Subramaniam et al. [17]	Population-based longitudinal survey	>18 years and over residing in India (*n* = 3000)	Not needed (8.1% of vaccine resistant respondents), dangerous, lack of trust
Syed et al. [18]	Cross-sectional study	>18 years, residing in Malaysia (*n* = 1411)	Not needed, dangerous (including fear of side-effects (95.8%, RII = 0.98)), lack of trust, cultural reasons
Wong et al. [19]	Cross-sectional survey	>18 years from 973 households in Soweto and 1442 households in Klerksdorp	Not needed (some participants believed that the influenza vaccine would not prevent influenza (Soweto: 23, 19%; Klerksdorp: 17, 19%; *p* = 0.9)), dangerous, lack of trust and cultural reasons

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
