# Peer review of "Causes of Vaccine Hesitancy in Adults for the Influenza and COVID-19 Vaccines: A Systematic Literature Review"

_vaccines, 2022, doi:10.3390/vaccines10091518_

Round 1

Reviewer 1 Report

The work done by Kumar et al entitled "Causes of Vaccine Hesitancy in Adults for the Influenza and 2 COVID-19 Vaccines: A Systematic Literature Review" needs a lot of revision before being accepted for publications.

The order of the review needs to be rearranged, for example, some sections needs to be re ordered.

the idea is week and not correlated to what is written well.

The authors need to revise the section of the lack of need for vaccination.

Table 1 is messy and could be a supplementary.

Author Response

Reviewer 1 Comment

Response

The order of the review needs to be rearranged, for example, some sections needs to be re ordered.

We agree that the numbering of the sections was confusing and have now renumbered them to make the paper more logical and easier to follow.

The authors need to revise the section of the lack of need for vaccination.

We have revised this section to improve the flow.

Table 1 is messy and could be a supplementary.

We have now reformatted the table so that is can be viewed on one page and is more readable.

Reviewer 2 Report

Kumar et al. reviewed on the causes of vaccine hesitancy for influenza and COVID-19 vaccines. Three main concerns were found to be the main cause of vaccine hesitancy. i) Vaccine safety, ii) lack of trust, lack of need for vaccination, iv) cultural reasons. The review is very interesting and I recommend its publication. However, we we can not ignore the real side effects of some COVID-19 vaccines. Please discuss the reported side effects after vaccination with COVID-19. Additionally, people refused to be vaccinated with certain COVID-19 due to the documented side effects. Please discuss? 

I suggest also to add the following recommendations in the discussion part: The following recommendations should be considered can help achieve higher public health and patient vaccination levels and less resistance to getting vaccinated. i) Talking with positive expectations, ii) Stating one’s practice procedures clearly, iii) Use active listening skills to hear the core of any patient doubts or fears, iv) Use motivational interviewing skills to correct misunderstanding and increase motivation for getting vaccinated v) Including health professionals is essential among those whose opinions and attitudes are monitored, given their influence on patients’ decisions, because they are also subject to uncertainty about COVID-19 vaccines (Doi: https://doi.org/10.51585/gjm.2021.2.0006)

Other Minor revision: Please see the attached file.

Author Response

Reviewer 2 Comment

Response

The review is very interesting and I recommend its publication.

We thank the reviewer for their  positive comments.

However, we  cannot ignore the real side effects of some COVID-19 vaccines. Please discuss the reported side effects after vaccination with COVID-19.

We have now discussed this in the implications for practice section.

Additionally, people refused to be vaccinated with certain COVID-19 due to the documented side effects. Please discuss?

People reported concern of potential side effects but interestingly none of the studies showed that these were as a result of documented side effects in the literature.  We have now discussed this further in the implications for practice section.  We agree with the reviewer that this is important point and thank them for their suggestion.

I suggest also to add the following recommendations in the discussion part: The following recommendations should be considered can help achieve higher public health and patient vaccination levels and less resistance to getting vaccinated. i) Talking with positive expectations, ii) Stating one’s practice procedures clearly, iii) Use active listening skills to hear the core of any patient doubts or fears, iv) Use motivational interviewing skills to correct misunderstanding and increase motivation for getting vaccinated v) Including health professionals is essential among those whose opinions and attitudes are monitored, given their influence on patients’ decisions, because they are also subject to uncertainty about COVID-19 vaccines (Doi: https://doi.org/10.51585/gjm.2021.2.0006)

We have now made reference to the study and approaches suggested by the reviewer.

Other Minor revision: Please see the attached file.

We have made the minor revisions suggested and thank the reviewer for pointing these out.

Reviewer 3 Report

Thank you for asking me to review this article. Highly transmittable infectious diseases are public health emergencies of international concern. There is still no definitive cure for some of those highly transmittable illness. Immunization and breaking the chain of infection is the only successful approach to mitigate its spread. However, sufficient vaccination coverage is conditioned by the people’s acceptance of these vaccines. Understanding the determinants of vaccination hesitancy is a useful tool for implementing strategic measures aimed at improving patient compliance with vaccination with particular reference to anti-Influenza and anti-COVID-19 vaccinations. In this context, aim of the paper under review is to synthesizing the findings of studies identifying causes of

vaccine hesitancy to the COVID-19 and influenza vaccines.

The subject under study is certainly important, especially in the historical period we are experiencing. The article presents interesting results but it must be further improved.

Title: it can be improved, highlight the object of the study.

Abstract. I encourage the authors to add more detail about their core contributions in the abstract.

Introduction: The authors should make clearer what is the gap in the literature that is filled with this study. The authors must better frame their study within the vast body of literature that also addressed the issue of acceptance of the vaccination in the adult population related to their level of knowledge concerning COVID-19 (refer to articles with DOI: https://doi.org/10.3390/ijerph182010872), although subsequently, the specific age group were not considered.

Methods: The review is interesting but it is not clear what the prevalence of vaccine hesitancy. The authors report that “A meta-analysis was not appropriate due the heterogeneity of the data, and a narra- 80 tive synthesis of the data was carried out, using thematic synthesis.” But why didn't the authors perform a meta-analysis to report the exact prevalence? I suggest that you perform this statistical technique or consult with experts in meta-analysis of proportions/prevalences.

Did the authors peer-reviewed the search strategy using PRESS. (https://www.cadth.ca/resources/finding-evidence/press).

Discussion: I also suggest expanding. Emphasize the contribution of the study to the literature. The discussion must be updated with the comparison and discussion regarding knowledge and acceptance, a paragraph should be added with a proper reference (see the above mentioned reference). The Authors should add more practical recommendations for the reader, based on their findings. Also, the section of limitations and future search is also very short, the Authors could elaborate on that.

Author Response

Reviewer 3 Comment

Response

The subject under study is certainly important, especially in the historical period we are experiencing. The article presents interesting results.

We thank the reviewer for their  positive comments

Title: it can be improved, highlight the object of the study.

We have reviewed the title ‘Causes of Vaccine Hesitancy in Adults for the Influenza and COVID-19 Vaccines: A Systematic Literature Review’ against the aim ‘This review aimed to fill this gap by synthesising the findings of studies identifying causes of vaccine hesitancy to the COVID-19 and influenza vaccines’ and feel that these are currently aligned.

Abstract. I encourage the authors to add more detail about their core contributions in the abstract.

We thank the reviewer for this suggestion and have edited the abstract as suggested.

Introduction: The authors should make clearer what is the gap in the literature that is filled with this study. 

We have now edited the introduction to make this clearer.

Introduction: The authors must better frame their study within the vast body of literature that also addressed the issue of acceptance of the vaccination in the adult population related to their level of knowledge concerning COVID-19 (refer to articles with DOI: https://doi.org/10.3390/ijerph182010872), although subsequently, the specific age group were not considered

We agree with the reviewer that the literature is vast.  We have therefore not referred to individual primary studies in the literature in our introduction, but rather to other reviews. 

Methods: The review is interesting but it is not clear what the prevalence of vaccine hesitancy. The authors report that “A meta-analysis was not appropriate due the heterogeneity of the data, and a narrative synthesis of the data was carried out, using thematic synthesis.” But why didn't the authors perform a meta-analysis to report the exact prevalence? I suggest that you perform this statistical technique or consult with experts in meta-analysis of proportions/prevalences.

The aim of our study was to synthesise literature related to the causes of vaccine hesitancy, rather than the prevalence.  In line with our aim, we did not include search terms related to the prevalence and therefore would therefore not have identified all studies reporting this.  We have now removed findings related to prevalence from the supplementary information to avoid any confusion.

Methods: Did the authors peer-reviewed the search strategy using PRESS. (https://www.cadth.ca/resources/finding-evidence/press).

We have checked our strategy against the PRESS guidance and now stated this in the paper.

Discussion: I also suggest expanding. Emphasize the contribution of the study to the literature. The discussion must be updated with the comparison and discussion regarding knowledge and acceptance, a paragraph should be added with a proper reference (see the above mentioned reference).

We had discussed the importance of information provision in terms of vaccine acceptance, referring to other reviews and highlighting what our review adds to these.  We referenced reviews rather than primary studies, as the literature in this area if vast.

Discussion: The Authors should add more practical recommendations for the reader, based on their findings.

We have now added an additional paragraph to the ‘recommendations for practice’ section.

Discussion: Also, the section of limitations and future search is also very short, the Authors could elaborate on that.

We have now expanded this section as suggested by the reviewer.

Round 2

Reviewer 1 Report

Thanks for improving

Author Response

Thank-you for your positive response.

Reviewer 2 Report

The manuscript is highly improved.

Minor comments. 

Line 260: "Covid 19" should be "COVID-19"

Line 262: Please delete " ]."

Author Response

Reviewer 2 Comment

Response

The manuscript is highly improved.

We thank the reviewer for their  positive comments.

Line 260: "Covid 19" should be "COVID-19"

Thank-you for pointing this out.  We have now amended accordingly.

Line 262: Please delete " ]."

Thank-you for pointing this out.  We have now amended accordingly

Reviewer 3 Report

Although the Authors tried to improve the manuscript, some of the concerns raised still remain and were not addressed. Therefore, the paper is not suitable to publication at this stage.

Author Response

Reviewer 3 Comment

Response

Although the Authors tried to improve the manuscript, some of the concerns raised still remain and were not addressed. Therefore, the paper is not suitable to publication at this stage.

We provided a point by point response to all the reviewers’ comments in our previous response.  We would be happy to address any further concerns that are raised.